# Towards Conceptual Compression

**Karol Gregor**
Google DeepMind
karolg@google.com

**Frederic Besse**
Google DeepMind
fbesse@google.com

**Danilo Jimenez Rezende**
Google DeepMind
danilor@google.com

**Ivo Danihelka**
Google DeepMind
danihelka@google.com

**Daan Wierstra**
Google DeepMind
wierstra@google.com

## Abstract

We introduce convolutional DRAW, a homogeneous deep generative model achieving state-of-the-art performance in latent variable image modeling. The algorithm naturally stratifies information into higher and lower level details, creating abstract features and as such addressing one of the fundamentally desired properties of representation learning. Furthermore, the hierarchical ordering of its latents creates the opportunity to selectively store global information about an image, yielding a high quality 'conceptual compression' framework.

## 1 Introduction

Deep generative models with latent variables can capture image information in a probabilistic manner to answer questions about structure and uncertainty. Such models can also be used for representation learning, and the associated procedures for inferring latent variables are vital to important application areas such as (semi-supervised) classification and compression.

In this paper we introduce convolutional DRAW, a new model in this class that is able to transform an image into a progression of increasingly detailed representations, ranging from global conceptual aspects to low level details (see Figure 1). It significantly improves upon earlier variational latent variable models (Kingma & Welling, 2014; Rezende et al., 2014; Gregor et al., 2014). Furthermore, it is simple and fully convolutional, and does not require complex design choices, just like the recently introduced DRAW architecture (Gregor et al., 2015). It provides an important insight into building good variational auto-encoder models of images: positioning multiple layers of stochastic variables 'close' to the pixels (in terms of nonlinear steps in the computational graph) can significantly improve generative performance. Lastly, the system's ability to stratify information has the side benefit of allowing it to perform high quality lossy compression, by selectively storing a higher level subset of inferred latent variables, while (re)generating the remainder during decompression (see Figure 3).

In the following we will first discuss variational auto-encoders and compression. The subsequent sections then describe the algorithm and present results both on generation quality and compression.

### 1.1 Variational Auto-Encoders

Numerous deep generative models have been developed recently, ranging from restricted and deep Boltzmann machines (Hinton & Salakhutdinov, 2006; Salakhutdinov & Hinton, 2009), generative adversarial networks (Goodfellow et al., 2014), autoregressive models (Larochelle & Murray, 2011; Gregor & LeCun, 2011; van den Oord et al., 2016) to variational auto-encoders (Kingma & Welling, 2014; Rezende et al., 2014; Gregor et al., 2014). In this paper we focus on the class of models in the

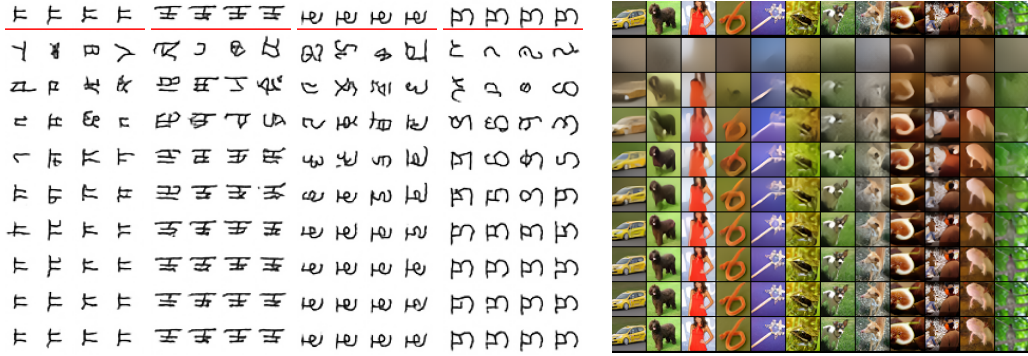

Figure 1: **Conceptual Compression**. The top rows show full reconstructions from the model for Omniglot and ImageNet, respectively. The subsequent rows were obtained by storing the first $t$ iteratively obtained groups of latent variables and then generating the remaining latents and visibles using the model (only a subset of all possible $t$ values are shown, in increasing order). **Left: Omniglot reconstructions**. Each group of four columns shows different samples at a given compression level. We see that the variations in the latter samples concentrate on small details, such as the precise placement of strokes. Reducing the number of stored bits tends to preserve the overall shape, but increases the symbol variation. Eventually a varied set of symbols is generated. Nevertheless even in the first row there is a clear difference between variations produced from a given symbol and those between different symbols. **Right: ImageNet reconstructions**. Here the latent variables were generated with zero variance (ie. the mean of the latent prior is used). Again the global structure is captured first and the details are filled in later on.

variational auto-encoding framework. Since we are also interested in compression, we present them from an information-theoretic perspective.

Variational auto-encoders consist of two neural networks: one that generates samples from latent variables ('imagination'), and one that infers latent variables from observations ('recognition'). The two networks share the latent variables. Intuitively speaking one might think of these variables as specifying, for a given image, at different levels of abstraction, whether a particular object such as a cat or a dog is present in the input, or perhaps what the exact position and intensity of an edge at a given location might be. During the recognition phase the network acquires information about the input and stores it in the latent variables, reducing their uncertainty. For example, at first not knowing whether a cat or a dog is present in the image, the network observes the input and becomes nearly certain that it is a cat. The reduction in uncertainty is quantitatively equal to the amount of information that the network acquired about the input. During generation the network starts with uncertain latent variables and samples their values from a prior distribution. Different choices will produce different visibles.

Variational auto-encoders provide a natural framework for unsupervised learning – we can build hierarchical networks with multiple layers of stochastic variables and expect that, after learning, the representations become more and more abstract for higher levels of the hierarchy. The pertinent questions then are: can such a framework indeed discover such representations both in principle and in practice, and what techniques are required for its satisfactory performance.

## 1.2 Conceptual Compression

Variational auto-encoders can not only be used for representation learning but also for compression. The training objective of variational auto-encoders is to compress the total amount of information needed to encode the input. They achieve this by using information-carrying latent variables that express what, before compression, was encoded using a larger amount of information in the input. The information in the layers and the remaining information in the input can be encoded in practice as explained later in this paper.

The achievable amount of lossless compression is bounded by the underlying entropy of the image distribution. Most image information as measured in bits is contained in the fine details of the image.

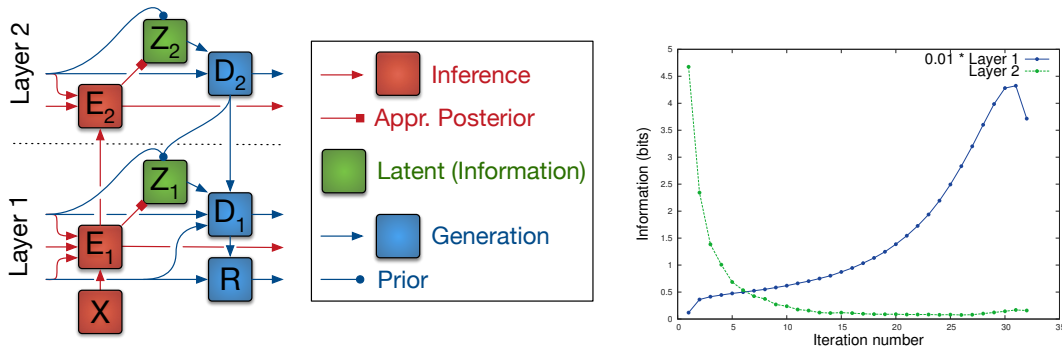

Figure 2: **Two-layer convolutional DRAW**. A schematic depiction of one time slice is shown on the left. $X$ and $R$ denote input and reconstruction, respectively. On the right, the amount of information at different layers and time steps is shown. A two-layer convolutional DRAW was trained on ImageNet, with a convolutional first layer and a fully connected second layer. The amount of information at a given layer and iteration is measured by the KL-divergence between the prior and the posterior (5). When presented with an image, first the top layer acquires information and then the second slowly increases, suggesting that the network first acquires 'conceptual' information about the image and only then encodes the remaining details. Note that this is an illustration of a two-layer system, whereas most experiments in this paper, unless otherwise stated, were performed with a one-layer version.

Thus we might reasonably expect that future improvements in lossless compression technology will be bounded in scope.

Lossy compression, on the other hand, holds much more potential for improvement. In this case the objective is to best compress an image in terms of quality of similarity to the original image, whilst allowing for some information loss. As an example, at a low level of compression (close to lossless compression), we could start by reducing pixel precision, e.g. from 8 bits to 7 bits. Then, as in JPEG, we could express a local 8x8 neighborhood in a discrete cosine transform basis and store only the most significant components. This way, instead of introducing quantization artefacts in the image that would appear if we kept decreasing pixel precision, we preserve higher level structures but to a lower level of precision. Nevertheless, if we want to improve upon this and push the limits of what is possible in compression, we need to be able to identify what the most salient 'aspects' of an image are.

If we wanted to compress images of cats and dogs down to one bit, what would that bit ideally represent? It is natural to argue that it should represent whether the image contains either a cat or a dog. How would we then produce an image from this single bit? If we have a good generative model, we can simply generate the entire image from this single latent variable by ancestral sampling, yielding an image of a cat if the bit corresponds to 'cat', and an image of a dog otherwise. Now let us imagine that instead of compressing down to one bit we wanted to compress down to ten bits. We can then store some other important properties of the animal as well – e.g. its type, color, and basic pose. Conditioned on this information, everything else can be probabilistically 'filled in' by the generative model during decompression. Increasing the number of stored bits further we can preserve more and more about the image, still filling in the fine pixel-level details such as precise hair structure, or the exact pattern of the floor, etc. Most bits indeed concern such low level details. We refer to this type of compression – compressing by preferentially storing the higher levels of representation while generating/filling-in the remainder – 'conceptual compression'.

Importantly, if we solve *deep* representation learning with latent variable generative models that generate high quality samples, we simultaneously achieve the objective of lossy compression mentioned above. We can see this as follows. Assume that the network has learned a hierarchy of progressively more abstract representations. Then, to get different levels of compression, we can store only the corresponding number of topmost layers and generate the rest. By solving unsupervised deep learning, the network would order information according to its importance and store it with that priority.

## 2 Convolutional DRAW

Below we present the equations for a one layer system (for a two layer system the reader is referred to the supplementary material):

For $t = 1, \ldots, T$                   At the end, at time T,

$$\epsilon_t = x - \mu(r_{t-1}) \tag{1}$$
$$h_t^e = \text{RNN}(x, \epsilon_t, h_{t-1}^e, h_{t-1}^d) \tag{2}$$
$$z_t \sim q_t = q(z_t | h_t^e) \tag{3}$$
$$p_t = p(z_t | h_{t-1}^d) \tag{4}$$
$$L_t^z = KL(q_t | p_t) \tag{5}$$
$$h_t^d = \text{RNN}(z_t, h_{t-1}^d, r_{t-1}) \tag{6}$$
$$r_t = r_{t-1} + W h_t^d \tag{7}$$

$$\mu, \alpha = \text{split}(r_T) \tag{8}$$
$$p^x = \mathcal{N}(\mu, \exp(\alpha))) \tag{9}$$
$$q^x = \mathcal{U}(x - s/2, x + s/2) \tag{10}$$
$$L^x = \log(q^x / p^x) \tag{11}$$
$$L = \beta L^x + \sum_{t=1}^{T} L_t^z \tag{12}$$

Long Short-Term Memory networks (LSTM; Hochreiter & Schmidhuber, 1997) are used as the recurrent modules (RNN) and convolutions are used for all linear operations. We follow the computations and explain them and the variables as we go along. The input image is $x$. The canvas variable $r_{t-1}$, initialized to a bias, carries information about the current reconstruction of the image: a mean $\mu(r_{t-1})$ and a log standard deviation $\alpha(r_{t-1})$. We compute the reconstruction error $\epsilon_t$. This, together with $x$, is fed to the encoder RNN (E in the diagram), which updates its internal state and produces an output vector $h_t^e$. This goes into the approximate posterior distribution $q_t$ from which $z_t$ is sampled. The prior distribution $p_t$ and the latent loss $L_t^z$ are calculated. $z_t$ is passed to the decoder and $L_t^z$ measures the amount of information about $x$ that is transmitted using $z_t$ to the decoder at this time. The decoder (D in the diagram) updates its state and outputs the vector $h_t^d$ which is then used to update the canvas $r_t$. At the end of the recurrence, the canvas consists of the values of $\mu$ and $\alpha = \log \sigma$ of the Gaussian distribution $p(x|z_1, \ldots, z_T)$ (or analogous parameters for other distributions). This probability is computed for the input $x$ as $p^x$. Because we use a real valued distribution, but the original data has 256 values per color channel for a typical image, we encode this discretization as a uniform distribution $\mathcal{U}(x - s/2, x + s/2)$ of width equal to the discretization $s$ (typically $1/255$) around $x$. The input cost is then $L^x = \log(q^x / p^x)$, it is always non-negative, and measures the number of bits (nats) needed to describe $x$ knowing $(z_1, \ldots, z_T)$. The final cost is the sum of the two costs $L = L^x + \sum_{t=1}^{T} L_t^z$ and equals the amount of information that the model uses to compress $x$ losslessly. This is the loss we use to report the likelihood bounds and is the standard loss for variational auto-encoders. However, we also include a constant $\beta$ and train models with $\beta \neq 1$ to observe the visual effect on generated data and to perform lossy compression as explained in section 3. Values $\beta < 1$ put less pressure on the network to reconstruct exact pixel details and increase its capacity to learn a better latent representation.

The general multi-layer architecture is summarized in Figure 2 (left). The algorithm is loosely inspired by the architecture of the visual cortex (Carlson et al., 2013). We will describe known cortical properties and in brackets the correspondences in our diagram. The visual cortex consists of hierarchically organized areas such as $V_1$, $V_2$, $V_4$, $IT$ (in our case: layers $1, 2, \ldots$). Each area such as $V_1$ is a composite structure consisting of six sublayers each most likely performing different functions (in our case: $E$ for encoding, $Z$ for sampling and information measuring, $D$ and $R$ for decoding). Eyes saccade around three times per second with blank periods in between. Thus the cortex has about 250ms to consider each input. When an input is received, there is a feed-forward computation that progresses to high levels of hierarchy such as $IT$ in about 100ms (in our case: the input is passed through the $E$ layers). The architecture is recurrent (our architecture as well) with a large amount of feedback from higher to lower layers (in our case: each $D$ feeds into the $E, Z, D, R$ layers of the next step), and can still perform significant computations before the next input is processed (in our case: the iterations of DRAW).

## 3 Compression Methodology

In this section we show how instances of the variational auto-encoder paradigm (including convolutional DRAW) can be turned into compression algorithms. Note however that storing subsets of

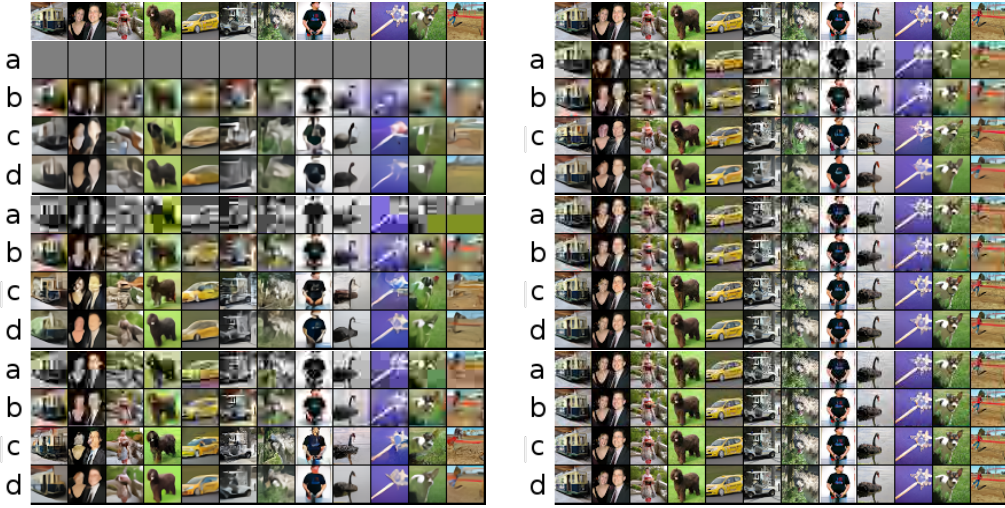

Figure 3: **Lossy Compression.** Example images for various methods and levels of compression. Top row: original images. Each subsequent block has four rows corresponding to four methods of compression: (a) JPEG, (b) JPEG2000, (c) convolutional DRAW with full prior variance for generation and (d) convolutional DRAW with zero prior variance. Each block corresponds to a different compression level; in order, the average number of bits per input dimension are: 0.05, 0.1, 0.15, 0.2, 0.4, 0.8 (bits per image: 153, 307, 460, 614, 1228, 2457). In the first block, JPEG was left gray because it does not compress to this level. Images are of size $32 \times 32$. See appendix for $64 \times 64$ images.

latents as described above results in good compression only if the network separates high level from low level information. It is not obvious whether this should occur to a satisfactory extent, or at all. In the following sections we will show that convolutional DRAW does in fact have this desirable property. It stratifies information into a progression of increasingly abstract features, allowing the resulting compression algorithm to select a degree of compression. What is appealing here is that this occurs naturally in such a simple homogeneous architecture.

The underlying compression mechanism is arithmetic coding (Witten et al., 1987). Arithmetic coding takes as input a sequence of discrete variables $x_1, \ldots, x_t$ and a set of probabilities $p(x_t | x_1, \ldots, x_{t-1})$ that predict the variable at time $t$ from the previous ones. It then compresses this sequence to $L = -\sum_t \log_2 p(x_t | x_1, \ldots, x_{t-1})$ bits plus a constant of order one.

We can use variational auto-encoders for compression as follows. First, train the model with an approximate posterior $q$ that has a variance independent from the input. After training, discretize the latent variables $z$ to the size of the variance of $q$. When compressing an input, assign $z$ to the nearest discretized point to the mean of $q$ instead of sampling from $q$. Calculate the discrete probabilities $p$ over the values of $z$. Retrain decoder and $p$ to perform well with the discretized values. Now, we can use arithmetic coding directly, having the probabilities over discrete values of $z$. This procedure might require tuning to achieve the best performance. However such process is likely to work since there is another, less practical way to compress that is guaranteed to achieve the theoretical value.

This second approach uses bits-back coding (Hinton & Van Camp, 1993). We explain only the basic idea here. First, discretize the latents down to a very high level of precision and use $p$ to transmit the information. Because the discretization precision is high, the probabilities for discrete values are easily assigned. That will preserve the information but it will cost many bits, namely $-\log_2 p^d(z)$ where $p^d$ is the prior under that discretization. Now, instead of choosing a random sample $z$ from the approximate posterior $q^d$ under the discretization when encoding, use another stream of bits that needs to be transmitted, to choose $z$, in effect encoding these bits into the choice of $z$. The encoded amount is $-\log_2 q^d(z)$ bits. When $z$ is recovered at the receiving end, both the information about the current input and the other information is recovered and thus the information needed to encode the

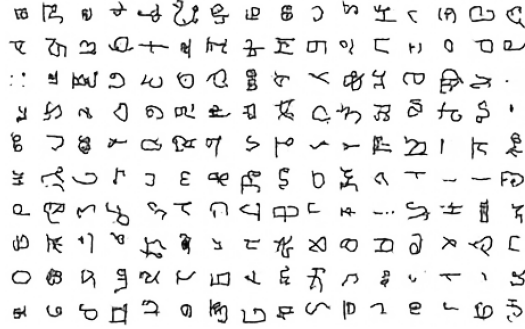

Figure 4: **Generated samples on Omniglot**.

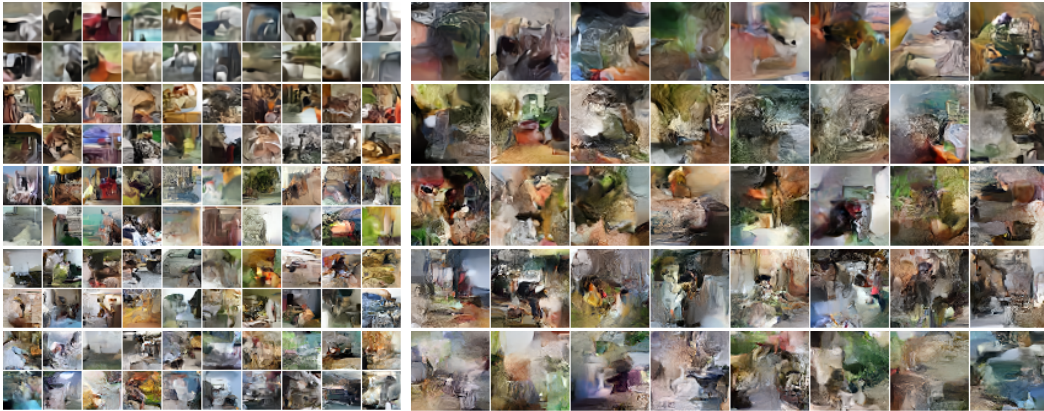

Figure 5: **Generated samples on ImageNet for different input cost scales.** On the left, $32 \times 32$ samples are shown with input cost $\beta$ in (12) equal to $\{0.2, 0.4, 0.6, 0.8, 1\}$ for each respective block of two rows. On the right, $64 \times 64$ are shown with input cost scale $\beta$ is $\{0.4, 0.5, 0.6, 0.8, 1\}$ for each row respectively. For smaller values of $\beta$ the network is less compelled to explain finer details of images, and produces 'cleaner' larger structures.

current input is $-\log_2 p^d(z) + \log_2 q^d(z) = -\log_2(p^d(z)/q^d(z))$. The expectation of this quantity is the KL-divergence in (5), which therefore measures the amount of information stored in a given latent layer. The disadvantage of this approach is that we need this extra data to encode a given input. However, this coding scheme works even if the variance of the approximate posterior is dependent on the input.

## 4 Results

All models (except otherwise specified) were single-layer, with the number of DRAW time steps $n_t = 32$, a kernel size of $5 \times 5$, and stride 2 convolutions between input layers and hidden layers with 12 latent feature maps. We trained the models on Cifar-10, Omniglot and ImageNet with 320, 160 and 160 LSTM feature maps, respectively. We use the version of ImageNet presented in (van den Oord et al., 2016). We train the network with Adam optimization (Kingma & Ba, 2014) with learning rate $5 \times 10^{-4}$. We found that the cost occasionally increased dramatically during training. This is probably due to the Gaussian nature of the distribution, when a given variable is produced too far from the mean relative to sigma. We observed this happening approximately once per run. To be able to keep training we store older parameters, detect such jumps and revert to the old parameters when they occur. In these instances training always continued unperturbed.

### 4.1 Modeling Quality

**Omniglot** The recently introduced Omniglot dataset Lake et al. (2015) is comprised of 1628 character classes drawn from multiple alphabets with just 20 samples per class. Referred to by some as the

'transpose of MNIST', it was designed to study conceptual representations and generative models in a low-data regime. Table 1 shows likelihoods of different models compared to ours. For our model, we only calculate the upper bound (variational bound) and therefore underestimate its quality. Samples generated by the model are shown in Figure 4.

**Cifar-10** Table 1 also shows reported likelihoods of different models on Cifar-10. Convolutional DRAW outperforms most previous models. The recently introduced Pixel RNN model (van den Oord et al., 2016) yields better likelihoods, but as it is not a latent variable model, it does not build representations, cannot be used for lossy compression, and is slow to sample from due to its autoregressive nature. At the same time, we must emphasize that the two approaches might be complementary, and could be combined by feeding the output of convolutional DRAW into the recurrent network of Pixel RNN.

We also show the likelihood for a (non-recurrent) variational auto-encoder that we obtained internally. We tested architectures with multiple layers, both deterministic and stochastic but with standard functional forms, and reported the best result that we were able to obtain. Convolutional DRAW performs significantly better.

**ImageNet** Additionaly, we trained on the version of ImageNet as prepared in (van den Oord et al., 2016) which was created with the aim of making a standardized dataset to test generative models. The results are in Table 1. Note that since this is a new dataset, few other methods have yet been applied to it.

In Figure 5 we show generations from the model. We trained networks with varying input cost scales as explained in the next section. The generations are sharp and contain many details, unlike previous versions of variational auto-encoder that tend to generate blurry images.

Table 1: **Test set performance of different models**. Results on $28 \times 28$ Omniglot are shown in *nats*, results on CIFAR-10 and ImageNet are shown in *bits/dim*. Training losses are shown in brackets.

| Omniglot | NLL |
|---|---|
| VAE (2 layers, 5 samples) | 106.31 |
| IWAE (2 layers, 50 samples) | 103.38 |
| RBM (500 hidden) | 100.46 |
| DRAW | $< 96.5$ |
| Conv DRAW | $< 92.0$ |

| ImageNet | NLL |
|---|---|
| Pixel RNN ($32 \times 32$) | 3.86 (3.83) |
| Pixel RNN ($64 \times 64$) | 3.63 (3.57) |
| Conv DRAW ($32 \times 32$) | 4.40 (4.35) |
| Conv DRAW ($64 \times 64$) | 4.10 (4.04) |

| CIFAR-10 | NLL |
|---|---|
| Uniform Distribution | 8.00 |
| Multivariate Gaussian | 4.70 |
| NICE [1] | 4.48 |
| Deep Diffusion [2] | 4.20 |
| Deep GMMs [3] | 4.00 |
| Pixel RNN [4] | 3.00 (2.93) |
| Deep VAE | $< 4.54$ |
| DRAW | $< 4.13$ |
| Conv DRAW | $< 3.58$ (3.57) |

## 4.2 Reconstruction vs Latent Cost Scaling

Each pixel (and color channel) of the data consists of 256 values, and as such, likelihood and lossless compression are well defined. When compressing the image there is much to be gained in capturing precise correlations between nearby pixels. There are a lot more bits in these low level details than in the higher level structure that we are actually interested in when learning higher level representations. The network might focus on these details, ignoring higher level structure.

One way to make it focus less on the details is to scale down the cost of the input relative to the latents, that is, setting $\beta < 1$ in (12). Generations for different cost scalings are shown in Figure 5, with the original objective being scale $\beta = 1$. Visually we can verify that lower scales indeed have a 'cleaner' high level structure. Scale 1 contains a lot of information at the precise pixel values and the network tries to capture that, while not being good enough to properly align details and produce real-looking patterns. Improving this might simply be a matter of network capacity and scaling: increasing layer size and depth, using more iterations, or using better functional forms.

### 4.3 Information Distribution

We look at how much information is contained at different levels and time steps. This information is simply the KL-divergence in (5) during inference. For a two layer system with one convolutional and one fully connected layer, this is shown in Figure 2 (right).

We see that the higher level contains information mainly at the beginning of computation, whereas the lower layer starts with low information which then gradually increases. This is desirable from a conceptual point of view. It suggests that the network first captures the overall structure of the image, and only then proceeds to 'explain' the details contained within that structure. Understanding the overall structure rapidly is also convenient if the algorithm needs to respond to observations in a timely manner. For the single layer system used in all other experiments, the information distribution is similar to the blue curve of Figure 2 (right). Thus, while the variables in the last set of iterations contain the most bits, they don't seem to visually affect the quality of reconstructed images to a large extent, as shown in Figure 1. This demonstrates the separation of information into global aspects that humans consider important from low level details.

### 4.4 Lossy Compression Results

We can compress an image lossily by storing only the subset of the latent variables associated with the earlier iterations of convolutional DRAW, namely those that encode the more high-level information about the image. The units not stored should be generated from the prior distribution (4). This amounts to decompression.

We can also generate a more likely image by lowering the variance of the prior Gaussian. We show generations with full variance in row 3 of each block of Figure 3 and with zero variance in row 4. We see that using the original variance, the network generates sharp details. Because the generative model is not perfect, the resulting images are less realistic looking as we lower the number of stored time steps. For zero variance we see that the network starts with rough details making a smooth image and then refines it with more time steps. All these generations are produced with a single-layer convolutional DRAW, and thus, despite being single-layer, it achieves some level of 'conceptual compression' by first capturing the global structure of the image and then focusing on details.

There is another dimension we can vary for lossy compression – the input scale introduced in subsection 4.2. Even if we store all the latent variables (but not the input bits), the reconstructed images will get less detailed as we scale down the input cost.

To build a high performing compressor, at each compression rate, we need to find which of the networks, input scales and number of time steps would produce visually good images. We have done the following. For several compression levels, we have looked at images produced by different methods and selected qualitatively which network gave the best looking images. We have not done this per image, just per compression level. We then display compressed images that we have not seen with this selection.

We compare our results to JPEG and JPEG2000 compression which we obtained using ImageMagick. We found however that these compressors were unable to produce reasonable results for small images ($3 \times 32 \times 32$) at high compression rates. Instead, we concatenated 100 images into one $3 \times 320 \times 320$ image, compressed that and extracted back the compressed small images. The number of bits per image reported is then the number of bits of this image divided by 100. This is actually unfair to our algorithm since any correlations between nearby images can be exploited. Nevertheless we show the comparison in Figure 3. Our algorithm shows better quality than JPEG and JPEG 2000 at all levels where a corruption is easily detectable. Note that even if our algorithm was trained on one specific image size, it can be used on arbitrarily sized images as it contains only convolutional operators.

## 5 Conclusion

In this paper we introduced convolutional DRAW, a state-of-the-art latent variable generative model which demonstrates the potential of sequential computation and recurrent neural networks in scaling up the performance of deep generative models. During inference, the algorithm arrives at a natural stratification of information, ranging from global aspects to low-level details. An interesting feature of the method is that, when we restrict ourselves to storing just the high level latent variables, we arrive at a 'conceptual compression' algorithm that rivals the quality of JPEG2000.

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
