[Supplementary Material]

# Towards Conceptual Compression
# Supplementary Material

**Karol Gregor**
Google DeepMind
karolg@google.com

**Frederic Besse**
Google DeepMind
fbesse@google.com

**Danilo Jimenez Rezende**
Google DeepMind
danilor@google.com

**Ivo Danihelka**
Google DeepMind
danihelka@google.com

**Daan Wierstra**
Google DeepMind
wierstra@google.com

## 1   Multi-layer Architectures

We explain how we can stack convolutional DRAW with a two layer example. The first layer is the same as the one we introduced in the main paper. The second layer has the same structure: recurrent encoder, recurrent decoder and a stochastic layer. The input to the second layer is the mean of the approximate posterior of the first layer. The output of the second layer biases the prior of the latent variable of the first layer and is also passed as input into the first layer decoder recurrent net. We don't use any reconstruction or error in the second layer. Systems with more layers can be built analogously. In the following equations, $e_i$ and $d_i$ refer to the encoder and decoder of the $i$-th layer respectively, and $z_t^i$ refers to the latent variable of the $i$-th layer at time step $t$.

For $t = 1, \dots, T$

$$
\begin{aligned}
\epsilon_t &= x - \mu(r_{t-1}) & (1)\\
h_t^{e_1} &= \text{RNN}(x, \epsilon_t, h_{t-1}^{e_1}, h_{t-1}^{d_1}) & (2)\\
\mu_t^1, \alpha_t^1 &= \text{split}(W^1 h_t^{e_1}) & (3)\\
z_t^1 &\sim q_t^1 = q(z_t^1 | \mu_t^1, \exp(\alpha_t^1)) & (4)\\
h_t^{e_2} &= \text{RNN}(h_t^{e_1}, h_{t-1}^{e_2}, h_{t-1}^{d_2}) & (5)\\
\mu_t^2, \alpha_t^2 &= \text{split}(W^2 h_t^{e_2}) & (6)\\
z_t^2 &\sim q_t^2 = q(z_t^2 | \mu_t^2, \exp(\alpha_t^2)) & (7)\\
h_t^{d_2} &= \text{RNN}(z_t^2, h_{t-1}^{d_2}) & (8)\\
h_t^{d_1} &= \text{RNN}(z_t^1, h_{t-1}^{d_1}, h_t^{d_2}, r_{t-1}) & (9)\\
r_t &= r_{t-1} + W h_t^{d_1} & (10)
\end{aligned}
$$

where $x$ is the input image, $r_t$ is the canvas at time $t$ consisting of a mean $\mu(r_t)$ and a log standard deviation $\alpha(r_t)$, $q_t^i$ is the approximate posterior of the $i$-th layer at time $t$, and $W^i$, $W$ are weights used in the $i$-th layer and in the canvas accumulation step respectively.

The latent losses are defined by:

$$
\begin{aligned}
p_t^2 &= p(z_t^2 | h_{t-1}^{d_2}) & (11)\\
p_t^1 &= p(z_t^1 | h_{t-1}^{d_1}, h_t^{d_2}) & (12)\\
L_t^{z_2} &= KL(q_t^2 | p_t^2) & (13)
\end{aligned}
$$

$$L_t^{z_1} \quad = \quad KL(q_t^1|p_t^1) \tag{14}$$

where $p_t^i$ is the prior of the $i$-th layer at time $t$, and $L_t^{z_i}$ is the latent loss of the $i$-th layer at time $t$.

At the end, at time T, the likelihood is calculated as:

$$\mu, \alpha \quad = \quad \text{split}(r_T) \tag{15}$$

$$p^x \quad = \quad \mathcal{N}(\mu, \exp(\alpha))) \tag{16}$$

$$q^x \quad = \quad \mathcal{U}(x - s/2, x + s/2) \tag{17}$$

$$L^x \quad = \quad \log(q^x/p^x) \tag{18}$$

$$L \quad = \quad \beta L^x + \sum_{t=1}^{T} L_t^{z_1} + \sum_{t=1}^{T} L_t^{z_2} \tag{19}$$

where $s$ is the quantization level of the image, typically $1/255$, $L^x$ is the pixel loss, $L$ is the final variational lower bound which we optimize for, and $\beta$ is a constant that modulates the amount of pressure we put on the network to reconstruct the image accurately at the pixel level.

## 2   Additional Samples

Below we show samples trained on $64 \times 64$ ImageNet with input cost scaling $\beta = 0.4$ (Figure 1) and $\beta = 1$ (Figure 2), as well as lossy compression results (Figure 3 and Figure 4).

Figure 1: Generated samples from a network trained on $64 \times 64$ ImageNet with input scaling $\beta = 0.4$.

Figure 2: Generated samples from a network trained on $64 \times 64$ ImageNet with input scaling $\beta = 1$.

Figure 3: **Lossy Compression, Part 1** Analogous to Figure 2 of the main paper but for $64 \times 64$ inputs. Example images for various methods and amounts of compression. Top block: original images. Each subsequent block has four methods of compression: JPEG, JPEG2000, convolutional DRAW with full prior variance for generation and convolutional DRAW with zero prior variance. Different blocks correspond to different compression levels, from top to bottom with bits per input dimension: 0.05, 0.1, 0.15, 0.2, 0.4, 0.8. In the first block, JPEG was left gray because it does not compress to this level.

Figure 4: **Lossy Compression, Part 2**