[Reviews · NeurIPS 2016]

Reviewer 1

Summary

This paper introduces a new approach to unsupervised representation learning for images based on recurrent neural nets. The main innovation is to alter the cost function to encourage modeling of high-level features rather than fine details. This leads to an effective framework for lossy compression of images.

Qualitative Assessment

This is obviously a high quality paper with solid technical contributions and impressive results. The main question for a paper like this one is whether it is award-worthy or not. This is a weighty question because awards function not just to highlight the best results, but also as exemplars of the highest standards in scientific research. From this point of view, I heartily support acceptance of this paper, but vote against a best paper award for reasons detailed below. Best features of this paper: 1. Compression To me, the focus on compression and the high level insights that emerge from this focus are the key feature of the paper. Research in unsupervised learning is often focused on perplexity/log likelihood scores to the exclusion of all else. While in principle these things are intimately related (through arithmetic coding, as the authors point out), the focus on a more operational task, lossy compression, reveals a different way of thinking about unsupervised learning. The same LL scores may, nevertheless, lead to drastically different results from the point of view of "conceptual compression". Even if this notion is only specified informally in this paper, I think it is a useful contribution to the field. It seems to me that compression has made a comeback (or never left?) with many papers talking about factorial codes as a motivation for unsupervised learning. (As in the recent InfoGAN paper, arxiv:1606.03657, which I mention again below.) The standard way of thinking is that in an independent or disentangled basis, (lossless) compression would become trivial. While that is an attractive notion, this paper points to a different path that doesn't appear to necessarily require any explicit disentangling. The point that very little information is contained at the conceptual level, with most contained at the detail level, is intuitive but difficult to express in the factorial coding point of view. What I find most exciting here is that there appears qualitatively to be a difference between building a latent representation based on the factorial idea (infoGAN) and the "conceptual" approach here. This makes for an exciting time for unsupervised learning. 2. Results I appreciate the effort to find and report on new standardized benchmarks that are more challenging than MNIST. I found the comparisons with JPEG compression compelling. I expect future developments will require equal parts of innovation for quantifying these types of comparisons, but I think the results here were enough to communicate a new and significant contribution. As a clarification here: the innovation is to use lossy compression in conjunction with a specific type of loss function to encourage and evaluate the learning of abstract details in unsupervised representation learning. There is certainly a great deal of existing work on lossy compression for images, and I think it would be appropriate to at least cite a review on that topic. The following critiques are, as mentioned above, geared towards justifying a high, but not top, score for this paper. 3. Reproducibility Compared to some disciplines, you could say that the work presented here is highly reproducible. On the other hand, computer science can be held to a higher standard and I believe it should be. I did not see any mention of code availability (even if the link would have to be omitted for anonymity). For comparison, a contemporaneous paper, the infoGAN paper, provides documented code with running examples that reproduce paper figures. Without any expertise implementing deep nets, I was able to get the code from that paper running in a few minutes. The implementation consisted of a few hundred lines of code, the main ingredients of which I could easily recognize. If, on the other hand, I had to reproduce this code based on the description in the paper, it would take, at best, months of work and, at worst, I would find that there were relevant details about initialization or hyper-parameters that were essential but incompletely described. For the long-term vitality of research in this field, I believe code availability is an essential consideration for top research, even if it is not (yet) a requirement for publishing. 4. Concepts I appreciated the overall framing of the paper and got a lot out of it. However, I am still dubious about the suggestion that the top latent factors specify a concept like "cat" or "dog", while the subsequent factors fill in detail. It looks to me more like the top factors specify "light, orange, oriented smudge", and the subsequent layers specify details that make it recognizable as a cat or a dog later. I would be surprised if there were consistently a "cat" neuron that appears in this framework, for instance. The flip side of the "concept" framing is that this model "learns to fill in detail later". It seems to me that the inclusion of convolutional structure is doing the heavy lifting here, as it does in most recent work, and is not an intrinsic advantage of the framework. I guess the test of this would be to see if this assertion continues to hold on non-image data. Regardless, the results are very interesting and further speculation and research on how and what is represented in this framework are warranted. p.s. The authors might want to check out this paper, http://arxiv.org/pdf/1503.03585.pdf. In particular they discuss various multi-scale approaches and the "dead leaf" images are a good test of this. It might be nice to nod in this direction and clarify how your work differs from these efforts to model images at multiple scales. edit: Based on the other reviews and the responses, it seems that more technical detail were omitted than I understood. I'm lowering my confidence and clarity score, but still argue for acceptance.

Confidence in this Review

1-Less confident (might not have understood significant parts)


Reviewer 2

Summary

The authors introduce convolutional DRAW, a version of a previously published model DRAW without attention. They illustrate the property of the model to effectively perform inference using a recurrent VAE and show that each step of this recurrence captures additional information about the global content of an image. In addition they study compression of the latent variables in order to reduce the codelength of images maximally. In their experiments, likelihoods for ConvDraw are reported beating several baselines without explicit spatial structure. In another experiment, the authors show conceptual compression by highlighting whioch layer encodes how muich information in a prelimninary plot.

Qualitative Assessment

The paper is a straightforward application of DRAW in a stationary setting without spatial components. The authors propose to study if the recurrence on the latents can learn to decompose interesting structure in the image in different timesteps and as such learn to compress an image 'conceptually'. The experiments in Figure 2 showing this is very preliminary and needs a lot more work to be made solid. In general, the paper, while interesting, would require more polish to make some of the authors claims to become more solid. On the positive side, the model clearly produces good samples and very good likelihoods. However, given that the technical novelty is limited and mostly the paper lives off of the clever application of previously published recurrent VAEs, studying exactly how the model appears to be achieving the results it does so would be interesting. As it is, the paper appears closer to a preliminary note with very encouraging initial results, but I would urge the authors to study the learning dynamics of the recurrence in more depth with respect to the conceptual decomposition of modeled content in terms of recurrence steps and in terms of model-depth. On a high level, it would be very interesting to see a model that can synthesize images by first sketching content on a high level and then gradually filling out details and this appears to be the authors' intuition here, but it would be worthwhile to invest more work in making this idea fully formed and clear.

Confidence in this Review

3-Expert (read the paper in detail, know the area, quite certain of my opinion)


Reviewer 3

Summary

This paper proposes a two-layer auto encoder model which can be utilized for samples generation and image compression.

Qualitative Assessment

1. This paper is not well prepared, there are considerable grammar and formatting mistakes. E.g., store select global information about an image (line 6), the amount of lossless compression one is able to achieve is (line 62), explanations of abbreviations in the abstract are not given. Additionally, there are many oral vocabularies in the paper. 2. The presentation of the introduction section is somewhat redundant, which needs to be refined. E.g., (line 66 - line 88). 3. Caption texts of figures are too long, try to remove them into the main body. Further, margins between figures and main body are relatively large. 4. Fcn.1-Fcn.11 present the convolutional DRAW model. Unfortunately, the presentation of this part is somewhat vague, definitions of some functions and variables are not given, e.g., p( ), q(), \mathcal{U}(), \mathcal{N}(), which makes it is very hard to follow the proposed model and problem. Moreover, in line 140 and 141, q (eq.3), p(eq.3), while Fcn.3 only has q. 5. Section 3 proposes the compression methodology, while there is not a clearly formulation of the problem. In line 131, two basic approaches for storing latent variables exist. However, the authors did not give a corresponding citation. Actually, reconstructing images by exploiting auto-encoder is not a tough task, the functionality of the introduction of the RNN in Fcn.5 is not obvious. 6. Fig.4 and Fig.5 show generated samples on some benchmark dataset, while it is not a novel technical contribution but the paper did not do a comparison with existing methods. 7.There are only 15 references in the paper, which is somewhat less. The authors should make a detailed investigation and update the presentation and the experiment.

Confidence in this Review

2-Confident (read it all; understood it all reasonably well)


Reviewer 4

Summary

This paper demonstrates that one of the best performing deep generative models, convolutional DRAW, can be used to achieve qualitatively much better lossy compression that standard JPEG and JPEG2000 compression. Convolutional DRAW is proposed here as a straightforward extension of DRAW and it is also shown to beat many previous *non-convolutional* benchmarks for deep generative models with tractable likelihood (or likelihood bound), but did not beat pixel RNNs (which performed substantially better than conv DRAW).

Qualitative Assessment

The visual quality of reconstruction after decompression at the same number of bits is really impressive, compared to JPEG and JPEG2000. This is clearly the main contribution of this paper. Extending DRAW to be convolutional is unsurprising and so is the fact that it works better than regular DRAW. Nonetheless, these comparisons are useful for future comparisons. The method used for compression is standard (arithmetic coding, combined with the bits-back trick to avoid paying for the compression of the part of the information in the latent variable that is already accounted for by the prior). In practice this method is probably useless for consumer applications because of the vastly greater computational time needed for compression and decompression (the latter is especially important in realistic applications). However, this is still scientifically important, as it may motivate other work implementing fast approximations that could be computationally feasible in practical applications. Since pixel RNNs performed much better than conv. DRAW in NLL, it seems logical that one should use them for compression. This also eliminates for the messy bits-back procedure... The authors should at least have a comment to that effect in the paper! My main concern about this paper is clarity. I found it very poorly written. However, I believe this is fixable. Many concepts are poorly explained, and if I had not known about most of them before reading the paper, I would have found it uncomprehensible. The main problem is that CONCEPTS ARE USED BEFORE BEING (POORLY) EXPLAINED. This is all over. You will find some examples below. Another concern I have is that the proposed model is not really deep in the usual sense of the word. Even the 2-level hierarchy in fig 2 may be misleading, because z2 is not used to generate z1 (nor is z1 ~ q(z1|x) used to obtain q(z2|x)). It's more like there really is only one latent level (z1,z2), with a particular form of encoder and decoder. I was also concerned by the fact that a single-level model was used in the all the experiments (and especially the compression and generation experiments), suggesting that the 2-level experiment of fig 2 did not work that well in practice. Is it the case? Please clarify. fig. 1 caption: what does it mean that latent variables are generated with zero variance? From p or from q? I suppose from q. I would say that the q-variance is set to 0, making the autoencoder deterministic. fig. 2 "iterations", which ones? RNN time steps or training iterations? l. 101: put _t indices to clarify which variables change with t which don't l. 107: don't put the equations one paragraph before the explanation about the variables they contain!!!! l. 131-134: difficult to parse before one has read the rest of the paper and digested it. l. 139-141: NOT CLEAR ENOUGH! l. 140: define "width of q" (variance?) l. 141: define precisely how the compression and decompression steps are performed. For this readership, which knows already about VAEs, it's much more important to make that crystal clear (e.g., how to use samples from q, how to use the value of p, and why it makes sense) than to re-explain VAEs. l. 142: why should discretization be independent of the input? l. 147: what does the following mean: "for the obtained likelihoods to be as good as those obtained through sampling" l. 154: why p^d? what is d? later you drop the ^d. Why? JUSTIFY!!! l. 156: explain better how "the choice of z" can be used to encode bits, as an alternative to sampling z from q. l. 164: define n_t!!!!

Confidence in this Review

3-Expert (read the paper in detail, know the area, quite certain of my opinion)


Reviewer 5

Summary

The paper proposes to use convolutional recurrent neural networks for lossy compression of natural images. A convolutional variant of DRAW (Gregor et al. 2015) is proposed as a generative model for natural iamges. The state-of-the-art performance in terms of log likelihood on a few benchmarks has been reported. With the proposed recurrent generative model, the authors argue that the model separates high level information from low level information and, at inference time, the model progressively adds more details into existing representations. This leads to a lossy compression algorithm by sending the first a few number of bits and using the model to infer the rest. The proposed compression method is qualitatively compared with standard image compression algorithms, i.e. JPEG and JPEG 2000.

Qualitative Assessment

The proposed idea of using probabilistic generative models for lossy image compression is very interesting. This demonstrates a new application of generative models of images. The phenomenon of "abstract to detail" modelling described by the authors is potentially very useful and can inspire research in other area. The proposed convolutional variant of DRAW produces the state-of-the-art performance on a few image benchmarks. My major concern is that most parts of the paper are written in the form of abstract descriptions. This leads to many ambiguities of the proposed methods and makes the method almost impossible to reproduce. For example, the paper does not explain how to extend the proposed convolutional variant of DRAW into two layers (in Figure 2) and the details of how the latent representations are encoded into binary codes (Section 3). An quantitative evaluation of the lossy compression performance with better baseline methods is missing. As the lossy compression is the main contribution of the paper, a quantiative evaluation (e.g. signal to noise ratio) is necessary. JPEG and JPEG 2000 are not good baselines because the quantization of JPEG is not optimized for the benchmark and is not designed for handling very low resolution images (32x32). A comparison with other generative model (e.g. convolutional variational auto-encoder) will be very interesting. A few detailed questions: 1. The definition of q^0(x) in Line 120 is incorrect. Is "x" the random variable of q^0(x) or the observed image intensity? 2. The definition of L^x in (10) is problematic, because, if the predicted intensity "x" falls out of the uniform distribution in (8), L^x is negative infinite. 3. In line 141, the equation reference should be "eq. 4" instead of "eq. 3". 4. The experiment setting in Section 4.3 needs to be explained in more detail. Does Figure 2 show the KL-divergence during the inference of an image? How is the initial hidden state h^e and h^d set for RNN?

Confidence in this Review

2-Confident (read it all; understood it all reasonably well)


Reviewer 6

Summary

The paper centers on adding additional prior structure for images (convolutions) into the DRAW model introduced by Gregor et al. They motivate their approach from a compression view point where the goal seems to be to allocate bits in order of human concepts. Their compression results are visually competitive and they evaluate the lower bound on several image sets.

Qualitative Assessment

The paper is a natural construction to study following DRAW. The visual results and motivation from the view of compression are nice, but I have some reservations about the writing and the seemingly ad-hoc parameter \beta. I found the model definition to be poor and the first panel of figure 2 to be unenlightening. I referred back to the DRAW paper whose notation in one of its central figure uses real distributions. The use of real distributions helps separates model from inference. Equations 1-11 are presented more like a piece of code than an intuitive construction of human concepts. The separation of model and inference also leads to my question with some of the results in table 1. We know DRAW can be improved using a better variational approximation (for example see The Variational Gaussian Process [ICLR, 2016]). Is this improvement orthogonal to Conv-DRAW? Can Conv-DRAW benefit from a richer approximating family? My second reservation lies in \beta? Is \beta needed because the underlying Conv DRAW model is misspecified in someway. It seems like a bandage to the model that requires external tuning.

Confidence in this Review

2-Confident (read it all; understood it all reasonably well)